# *Staphylococcus saccharolyticus* Associated with Prosthetic Joint Infections: Clinical Features and Genomic Characteristics

**DOI:** 10.3390/pathogens10040397

**Published:** 2021-03-26

**Authors:** Bo Söderquist, Mastaneh Afshar, Anja Poehlein, Holger Brüggemann

**Affiliations:** 1School of Medical Sciences, Faculty of Medicine and Health, Örebro University, 701 82 Örebro, Sweden; 2Department of Laboratory Medicine, Clinical Microbiology, Örebro University Hospital, 701 85 Örebro, Sweden; 3Department of Biomedicine, Aarhus University, 8000 Aarhus, Denmark; m.afshar@biomed.au.dk (M.A.); brueggemann@biomed.au.dk (H.B.); 4Department of Genomic and Applied Microbiology, Institute of Microbiology and Genetics, University of Göttingen, 37077 Göttingen, Germany; apoehle3@gwdg.de

**Keywords:** *Staphylococcus saccharolyticus*, coagulase-negative staphylococci, whole-genome sequencing, prosthetic joint infections

## Abstract

The anaerobic coagulase-negative staphylococcal species *Staphylococcus saccharolyticus* is a member of the normal skin microbiota. However, *S. saccharolyticus* is rarely found in clinical specimens and its pathogenic potential is unclear. The clinical data of prosthetic hip (n = 5) and shoulder (n = 2) joint implant-associated infections where *S. saccharolyticus* was detected in periprosthetic tissue specimens are described. The prosthetic hip joint infection cases presented as “aseptic” loosening and may represent chronic, insidious, low-grade prosthetic joint infections (PJIs), eventually resulting in loosening of prosthetic components. All cases were subjected to one-stage revision surgery and the long-term outcome was good. The shoulder joint infections had an acute onset. Polymicrobial growth, in all cases with *Cutibacterium acnes*, was found in 4/7 patients. All but one case were treated with long-term administration of beta-lactam antibiotics. Whole-genome sequencing (WGS) of the isolates was performed and potential virulence traits were identified. WGS could distinguish two phylogenetic clades (clades 1 and 2), which likely represent distinct subspecies of *S. saccharolyticus*. Little strain individuality was observed among strains from the same clade. Strains of clade 2 were exclusively associated with hip PJIs, whereas clade 1 strains originated from shoulder PJIs. It is possible that strains of the two clades colonize different skin habitats. In conclusion, *S. saccharolyticus* has the potential to cause PJIs that were previously regarded as aseptic loosening of prosthetic joint devices.

## 1. Introduction

The anaerobic coagulase-negative staphylococcal species *Staphylococcus saccharolyticus* can be found on human skin, where it is much more abundant than previously recognized and may represent an important part of the normal skin microbiota [1,2]. *S. saccharolyticus* is rarely found in clinical specimens and its pathogenic potential is unclear. In a previous study, a few potential virulence traits of *S. saccharolyticus* strains isolated from blood cultures and prosthetic joint infections (PJIs) were found by whole-genome sequencing (WGS) and secretome analyses, including urease and hyaluronidase activities, the arginine deiminase pathway involved in pH regulation, and putative immunomodulatory proteins (e.g., immunodominant staphylococcal surface antigens A and B) [3]. In addition, toxins of the phenol-soluble modulin (PSM) family as well as virulence-associated regulatory systems such as the agr quorum-sensing system and the SrrAB two-component system were found.

*S. saccharolyticus* has rarely been reported to be associated with human infections. In the literature, approximately 10 clinical cases associated with or caused by *S. saccharolyticus* have been reported. These include three cases of spondylodiscitis [4,5,6] and two cases of infectious endocarditis—one case of native [7] and one case of prosthetic valve endocarditis [8]. Other cases include pyomyositis [9], pneumonia [10], empyema [11], and a case with a bone marrow infection [12]. The organism was also detected in blood cultures of 17 inpatients in an evident hospital outbreak of bacteremia in Germany [13]. Besides the spondylodiscitis cases, there is also a report describing a patient with a low-grade infection of the shoulder joint due to *S. saccharolyticus* following an intra-articular injection; *S. saccharolyticus* was isolated in pure culture in multiple samples (4/4 tissue specimens) [14]. Beyond that, there are no other reported cases of bacterial arthritis including foreign body infections such as PJIs. In a systematic literature review of PJIs associated with anaerobic bacteria, there were no reports of cases caused by *S. saccharolyticus* [15].

We have previously reported the WGS data of isolates of *S. saccharolyticus* obtained from blood cultures and PJIs [3]. Here we describe the clinical data of a total of seven PJI cases where *S. saccharolyticus* was isolated in association with a prosthetic joint device and provide further insight into the genomes and the population structure of disease-associated *S. saccharolyticus* isolates.

## 2. Results

### 2.1. Clinical Data

Over a five-year period (2013–2017), we identified five patients with loosening of components of hip joint prostheses where *S. saccharolyticus* was isolated from tissue samples collected during exchange surgery. There were no clinical signs of PJI in these cases. In addition, two cases of early PJIs due to *S. saccharolyticus* following shoulder arthroplasty surgery were identified.

Clinical data extracted from the patients’ records are presented in Table 1. The five hip arthroplasty (THA) patients were all male, and the median age was 75 years (range 65–76 years). The two patients with prosthetic shoulder infections were also male and were 72 and 49 years old, respectively (Table 1).

Four of the five THA patients showed a history of long duration of good functional status (range 7–16 years). They reported gradually increasing pain for months or even years until diagnostic imaging revealed loosening of the prosthetic devices, mainly the cup. In one case, the loosening presented only 14 months after primary arthroplasty surgery. Besides pain, no signs of inflammation or fever were noted. No sinus tracts were present. The median C-reactive protein (CRP) at diagnosis of loosening was <4 mg/L (range <4–5.6 mg/L) and median erythrocyte sedimentation rate (ESR) was 14 mm/1 h (range 3–28 mm/1 h). Two patients had underlying diseases: prostate cancer and insulin-treated diabetes mellitus, respectively.

In three out of five cases, the cultures from the perioperative tissue samples showed monomicrobial growth of *S. saccharolyticus*. In two cases, there was polymicrobial growth, in both cases a mixture with *Cutibacterium acnes* (Table 1). In one case only, 1/5 tissue samples showed growth of *S. saccharolyticus*, and thus the case did not fulfill the diagnostic criteria of a PJI [16].

In all the cases with loosening of the prosthetic device, there was no suspicion of a PJI at revision surgery, but tissue cultures were routinely taken. According to the microbiological results and reports, a decision was made not to initiate any antibiotic treatment in two cases. These cases were followed up on regarding their clinical status and laboratory parameters. The remaining three patients were prescribed oral amoxicillin (Table 1).

At follow-up, at least one year after one-stage revision surgery, the outcome was good to excellent in 4/5 cases and no walking aids were necessary (Table 1). No signs of relapse of a PJI were noted within a year in any of the cases.

The two cases of shoulder joint implant-associated infections were different in that they presented with an acute onset. The first case was a 72-year-old male who had a hemi-prosthesis inserted 9 days after a complicated proximal fracture of the right humerus. The patient experienced pain, swelling and redness of the shoulder less than three weeks post-operation. Ten days later, a soft-tissue revision and irrigation were performed. CRP was 73 mg/L at admission and per-operative tissue cultures displayed growth of *S. saccharolyticus* in 5/5 samples and *C. acnes* in 4/5 samples. The patient was treated with phenoxymethylpenicillin for 4 months. At long-term follow-up, the range-of-motion (ROM) was limited to 30°. The infection was considered to be cured.

The second case was a 49-year-old male with a complicated history of secondary osteoarthritis as a result of a native septic arthritis of the shoulder due to *Staphylococcus aureus* that was treated with isoxazolyl penicillin for three months. Almost one year after discontinuation of the antimicrobial treatment, a total arthroplasty shoulder surgery was performed. CRP was <4 mg/L. Per-operative tissue biopsies showed growth of *S. saccharolyticus* in 3/5 samples and *C. acnes* in 1/5. Clindamycin was administered immediately post-operatively and later replaced by oral treatment with phenoxymethylpenicillin for three months. However, after antibiotic treatment was discontinued, the patient noted increasing pain from the shoulder; the CRP remained low at <4 mg/L. Tissue biopsies were again obtained via a surgical incision performed in the operating theater. They showed growth of *S. saccharolyticus* in 2/6 samples and *C. acnes* in 1/6 samples. Treatment with amoxicillin was then re-instituted, and one month later, explantation of the prosthetic device was performed. All tissue biopsies obtained in that surgical procedure showed no growth. Treatment with amoxicillin with 1 g t.i.d. was given for three months. A shoulder prosthesis was reimplanted five months after extraction of the prosthetic device. Perioperative cultures at that operation showed no growth. A follow-up (>12 months later) showed good functional status of the shoulder.

### 2.2. Antibiotic Susceptibility Testing

The results of the antibiotic susceptibility tests are given in Table 2. All isolates were fully susceptible to all tested antibiotics including penicillin G and clindamycin but intrinsically resistant to metronidazole.

### 2.3. Genome Sequencing of Clinical S. saccharolyticus Isolates

A few *S. saccharolyticus* strains were previously sequenced, including strains DVP2-17-2406 and DVP5-16-4677 (both from hip PJIs, patients 1 and 2, respectively) and strain 13T028 (from shoulder PJI, patient 6) [3] (Table 3). Recently, the genomes of strains DVP5-16-4677 and 13T028 were completely closed [17]. Five new strains were sequenced in this study in order to have a more complete picture of PJI-associated *S. saccharolyticus* strains (Table 3). Now, at least one strain per patient has been sequenced. For one patient (patient 7), two strains were sequenced that were isolated four months apart. All sequenced genomes have a similar GC content (32%) and the genome size varied between 2321 kb and 2396 kb.

#### 2.3.1. Population Structure Based on Core-Genome Comparison

A Parsnp analysis was carried out to identify the core genome of *S. saccharolyticus* and to analyze the phylogeny, based on core-genome-located single-nucleotide polymorphisms (SNPs). The phylogenetic analysis divides the *S. saccharolyticus* population into two clades, clades 1 and 2, as previously noted [3] (Figure 1). Genomes of the two clades have an average nucleotide identity (ANI) of 97.9%. Thus, the two clades can be regarded as two distinct subspecies of *S. saccharolyticus*. Interestingly, all hip PJI isolates belong to clade 2, and all shoulder PJI isolates belong to clade 1 (Figure 1A). The latter clade also contains two blood culture isolates and the type strain of the species, NCTC 11807, a strain isolated from human blood. The core genome identified by Parsnp covers 93% of the closed reference genome (here: DVP5-16-4677). In total, 34,107 SNPs were detected in the population (Figure 1B). When just comparing clade 1 strains, 298 SNPs were detected in the core genome (strain 13T028 as reference), and when just comparing clade 2 strains, 645 SNPs were detected (strain DVP5-16-4677 as reference) (Appendix A). This low number of SNPs among strains of each clade indicates relatively little strain individuality. In contrast, clade 1 and clade 2 differed by 33,164 SNPs; these were distributed throughout the genome and affected all genes, underlining that clade 1 and clade 2 represent two separate subspecies (Figure 1B). We sequenced two isolates of patient 7: the strains DVP4-16-6166 and DVP1-17-1678 were isolated four months apart. Interestingly, no differences, i.e., no core-genome SNPs nor any difference in the accessory genome, could be identified in these strains (data not shown).

#### 2.3.2. Genome-Wide Synteny with a Small Accessory Genome

The genomes of *S. saccharolyticus* showed a strong synteny when comparing strains from the same clade, but also when comparing strains belonging to the two different clades (Figure 2). Only very few genomic islands were found. A list of clade-specific genes is given in Appendix A, based on the comparison of the two closed genomes of the clade 1 strain 13T028 and the clade 2 strain DVP5-16-4677. Clade 1 strains harbor two clade-specific islands >8 kb. The biggest island represents a 19 kb plasmid that harbors, among several genes encoding hypothetical proteins, a gene cluster putatively involved in bacteriocin biosynthesis and a toxin–antitoxin system. The other clade 1-specific island encodes a restriction modification system. Clade 2 strains harbor three clade-specific islands >8 kb, harboring around 60 genes in total (Appendix A). Two clade 2 strains (DVP5-16-4677 and 14T637) possess a 55 kb plasmid that harbors, among others, genes putatively involved in bacteriocin biosynthesis. We noticed that many clade-specific genes were frameshifted (34% in clade 1 and 40% in clade 2).

#### 2.3.3. Many Pseudogenes in the Genome of *S. saccharolyticus*

Striking is the presence of many pseudogenes in the genomes of *S. saccharolyticus*. Clade 1 and clade 2 strains carry on average 470 and 490 pseudogenes, respectively, as judged from the PGAP annotation pipeline. Thus, about 20% of all predicted genes are inactivated in each genome, mainly due to frameshift mutations. For the closed genomes of strains 13T028 and DVP5-16-4677, 469 and 493 genes were predicated to be non-functional. An analysis of the metabolic pathways that were affected by this genome decay was carried out using BlastKOALA. The tool could map 39% (182/469 genes) and 40% (198/493 genes) of the genes affected by mutational inactivation to pathways, respectively. The analysis showed that many different pathways were inactivated (Appendix A). Among these, amino acid biosynthesis pathways were strongly affected (Figure 3). According to this analysis, strain DVP5-16-4677 is not able to synthesize any amino acid, thus itis entirely auxotroph for all amino acids. Also, strain 13T028 is auxotroph for most amino acids, with tryptophan as a possible exception. In addition, strain 13T028 is predicted to be able to convert more amino acids into others, compared to strain DVP5-16-4677, for example, regarding the conversion of methionine to cysteine.

## 3. Discussion

In the present study, we reported seven male patients where *S. saccharolyticus* was isolated in association with a prosthetic joint device, five cases with loosening of the prosthetic device and two cases with an early postinterventional infection.

The five patients with loosening of the prosthetic joint devices were all total hip replacements where the acetabular cup was loose. In one case, a Lubinus prosthesis, the femoral stem was also loose, definitely indicating a true infection. All patients underwent one-stage exchange surgical procedures. Pre-operatively there were no suspicions of a PJI in any of the cases; thus, they were regarded as aseptic loosening of the cup. However, routine tissue biopsies showed growth of *S. saccharolyticus*, in three out of five cases in pure culture.

In two additional cases, the presentation was early manifestations with signs of an acute infection after primary surgery with shoulder arthroplasty. These two infections were polymicrobial. The two polymicrobial cases of hip infections also presented a mixture of two anaerobes, besides *S. saccharolyticus* and *C. acnes*. It is possible that these two skin commensals share the same niche and have similar growth behavior and nutrient requirements. Like *C. acnes*, *S. saccharolyticus* prefers anaerobic conditions and secretes multiple lipases, and thus has lipolytic activity [3].

In polymicrobial PJIs, *C. acnes* has been reported as a rare co-pathogen to coagulase-negative staphylococci such as *Staphylococcus epidermidis* and *Staphylococcus warneri* [18]. However, co-pathogens to *C. acnes* were not reported in the review of anaerobic PJIs by Shah et al. [15].

Co-morbidity was present in only two of our cases; diabetes mellitus and prostate cancer. This is in accordance with the review of Trojani et al. [6], where only three out of nine previously reported cases displayed any host risk factors. According to the same review, three reported cases had affected bone and joint (i.e., spondylodiscitis), and two had been associated with foreign body materials. Both these patients had prosthetic valve endocarditis.

All five THA patients presented with an “aseptic loosening” of the acetabular cup. The time period from implantation to loosening of the cup was long, in most cases many years. *S. saccharolyticus* could have been introduced during the primary surgical procedure, and thus represent a true low-virulent pathogen resulting in a low-grade, insidious infection eventually resulting in bone destruction and finally loosening of the prosthetic device. However, since *S. saccharolyticus* may be an overlooked and common part of the skin microbiome [2], it could also have been translocated more recently from breaches of the skin or wounds to the implant per continuitatem spread or by the hematogenous route.

All isolates of *S. saccharolyticus* were highly susceptible to beta-lactam antibiotics, such as benzylpenicillin, amoxicillin and carbapenems, and clindamycin and were also susceptible to vancomycin and chloramphenicol but intrinsic resistant to metronidazole according to the break-points defined by EUCAST (https://www.eucast.org/fileadmin/src/media/PDFs/EUCAST_files/Breakpoint_tables/v_11.0_Breakpoint_Tables.pdf; accessed on 18 March 2021). The optimal treatment for PJI caused by slow-growing anaerobic Gram-positive bacteria has not been defined, although the Swedish national guidelines for bone and joint infections recommend amoxicillin for oral follow-up treatment of PJI caused by *C. acnes*. However, combination therapy with, for example, rifampin and moxifloxacin, has been proposed. In the present study, five patients were successfully treated with amoxicillin or phenoxymethylpenicillin. Amoxicillin would be preferable due to its higher bioavailability. In two cases, a one-stage surgical revision procedure was performed and no follow-up treatment with antibiotics was done. Clinical follow-up showed good-to-excellent functional status. A meticulous revision when performing a one-stage exchange procedure may be sufficient to eradicate this low-grade infection that probably displays a low bacterial burden.

Regarding perioperative antibiotic prophylaxis in implant surgery where anaerobes such as *C. acnes* may cause infectious complications, e.g., shoulder arthroplasty surgery, addition of benzylpenicillin or clindamycin to cloxacillin (recommended as antibiotic prophylaxis for all primary joint replacements in Sweden [PRISS Expert Group 2; http://lof.se/patientsakerhet/vara-projekt/rekommendationer/]) (accessed on 18 March 2021) may be considered. However, the MIC values for cloxacillin of the *S. saccharolyticus* isolates were all <0.016 mg/L (data not shown).

All patients in the present study were men. The question of whether there is a sex difference regarding the relative abundance and distribution of *S. saccharolyticus* on the skin has not been addressed so far. However, a highly significant difference between men and women regarding *C. acnes* has been reported in the number of positive cultures from the chest prior to cardiac surgery [19]. In addition, following disinfection with 0.5% chlorhexidine in 70% alcohol, growth of *C. acnes* was still found in half of the patients. It is not known whether routine perioperative disinfection procedures are ineffective in eradicating or reducing the number of *S. saccharolyticus* on the skin. However, it seems likely due to the preference of the bacterium for anoxic conditions, which are found in deeper layers of the epidermis.

The genome sequences of the five PJI isolates sequenced in this study have extended the knowledge about the species *S. saccharolyticus*. The species can be divided into two phylogenetically distinct clades that can be regarded as distinct subspecies, given the substantial difference of the ANI between the two clades, with over 33,000 core-genome SNPs that were distributed throughout the genome. As a potentially interesting finding, the two clades seem to be associated with different body sites: all hip PJIs reported here are associated with clade 2 strains, whereas shoulder PJI isolates are clade 1 strains. Assuming a close proximity of bacterial habitat and infection site, this could indicate that different skin sites are colonized with phylogenetically different *S. saccharolyticus* types. However, due to the low sample size, this finding is preliminary and needs to be challenged in a future study with a higher sample size. If true, it could be related to niche-specific adaption of the two types of *S. saccharolyticus*.

The two clades of *S. saccharolyticus* differ in several regards. We found differences in the accessory genome, with clade 1- and clade 2-specific regions (Figure 2, Appendix A). Specific plasmids were found to be present in clade 1 and clade 2 strains, respectively. These encode many unknown functions, but larger gene clusters seem to be involved in the biosynthesis of clade-specific bacteriocins. Experimental evidence needs to be obtained in order to investigate the contribution of the accessory genome to clade-specific properties. Besides the accessory genome, the core genome of clade 1 and clade 2 strains harbors over 33,000 SNPs. Thus, many protein functions could be affected, e.g., by nonsynonymous substitutions. For example, we previously noticed that the hyaluronidase activity between the two clades differed: clade 1 strains had a stronger activity compared to clade 2 strains, as judged from a plate assay; SNPs were found in the hyaluronidase encoding gene and its promoter, which could explain the different activities between clade 1 and clade 2 strains [3]. Moreover, the high number of pseudogenes, indicative of recent and ongoing evolution, can affect clade 1 and clade 2 strains differently. As an example, we found that amino acid biosynthesis pathways are strongly affected by mutational inactivation in both clades (Figure 3). However, clade 2 strains seem to have accumulated more pseudogenes than clade 1 strains, affecting more reactions in amino acid biosynthesis pathways. More analyses and experimental verifications need to be done to explore the extent of genome decay in clade 1 and clade 2 strains and the functional consequences. Overall, clade 1 and clade 2 strains carry on average 470 and 490 pseudogenes, respectively. Thus, about 21% of all coding sequences encoded in the genome of *S. saccharolyticus* are inactivated, which might explain its slow growth and the preference for anaerobic conditions. This massive genome decay suggests that *S. saccharolyticus* has relatively recently changed habitats. The skin habitat on humans, possibly within deeper layers of the epidermis, likely provides anaerobic conditions, nutrients, cofactors and amino acids. A future study needs to investigate the core metabolism and nutritional requirements of *S. saccharolyticus*.

The virulence potential of *S. saccharolyticus* has so far not been investigated in detail. Several putative virulence factors have been found to be encoded in the genome, including hyaluronate lyase and urease [3]. In addition, other putative virulence factors were found such as a protein similar to BrkB of *Bordetella pertussis*, a protein essential for resistance to complement-dependent killing [20]. In addition, the microorganism has previously been shown to secrete an arsenal of factors with immune-stimulatory potential, such as heat shock proteins and immunodominant staphylococcal surface antigens A (IsaA) and B (IsaB) [3].

In conclusion, *S. saccharolyticus* may cause chronic, insidious, low-grade PJIs, eventually resulting in loosening of prosthetic devices such as the acetabular cup of a THA. In addition, *S. saccharolyticus* may also be involved in acute postinterventional infections of prosthetic devices, at least as part of polymicrobial infections accompanying *C. acnes*.

## 4. Materials and Methods

### 4.1. Patients

Bacterial isolates from PJIs are per routine stored and frozen at −80 °C at the Department of Laboratory Medicine, Clinical Microbiology, Örebro University Hospital. Since 2004, national guidelines have recommended that at least five per-operative samples should be taken when performing a surgical procedure such as DAIR (debridement, antibiotics and implant retention) of a suspected or verified deep orthopedic infection. It has become routine at our hospital to also collect multiple tissue samples when performing exchange arthroplasty surgery of prosthetic devices due to loosening. We therefore searched our database at the Department of Laboratory Medicine for all samples where ≥3 samples had been retrieved on the same occasion. All cases displaying isolates of *S. saccharolyticus* or anaerobic Gram-positive cocci were chosen for further analysis.

### 4.2. Bacterial Isolates

The isolates that were per routine identified as anaerobic staphylococci were assigned to species level by API 20 A (bioMérieux, Marcy-l’Etoile, France) before the implementation of matrix-assisted laser desorption/ionization time-of-flight mass spectrometry (MALDI-TOF MS) with a Microflex LT and Biotyper 3.1 (Bruker Daltonik, Bremen, Germany) in January 2014. All isolates were retrospectively assigned to species level by MALDI-TOF MS. Isolates were stored in preservation medium (trypticase soy broth with 0.3% yeast extract and 29% horse serum) at −80 °C. The identified isolates were subcultured on FAA plates (4.6% LAB 90 Fastidious Anaerobe Agar, LAB M, Heywood, United Kingdom) supplemented with 5% horse blood (*v*/*v*) and incubated at 36 °C in an anaerobic atmosphere for 2–5 days.

### 4.3. Antibiotic Susceptibility Testing

Minimum inhibitory concentration (MIC) was determined by Etest (bioMérieux, Marcy-l’Etoile, France) according to EUCAST guidelines (https://www.eucast.org/ast_of_bacteria/; accessed on 18 March 2021). Antibiotic susceptibility testing was performed on FAA plates with 0.5 McFarland suspensions of bacteria in NaCl and incubated at 36 °C in anaerobic conditions for 2 days.

### 4.4. Clinical Data

All isolates that could be identified as *S. saccharolyticus* by MALDI-TOF MS were linked to the patient’s ID (their unique Swedish personal number). The patients’ medical records were reviewed according to a predefined protocol in order to extract relevant clinical data.

C-reactive protein was determined at the Department of Laboratory Medicine, Clinical Chemistry; Örebro University Hospital according to routine method (ADVIA Chemistry AXP System, Siemens, Erlangen, Germany).

### 4.5. Genome Sequencing of S. saccharolyticus Isolates

Five *S. saccharolyticus* strains were selected for whole-genome sequencing (DVP1-17-2344, DVP4-16-6166, 14T0637, 13T098, DVP1-17-1678). Genomic DNA was isolated using the MasterPure Gram-positive DNA Purification Kit (EpiCentre MGP04100) according to the manufacturer’s instructions. The purity and quality of the gDNA were assessed on a 1% agarose gel and with a nanodrop apparatus (Thermo Fisher Scientific). The validation criteria were: sample quality 260/280 nm = 1.8–2 and 260/230 nm = 2.0–2.2 and a minimum concentration 2.5 µg/µL. The extracted DNA was used to generate Illumina shotgun paired-end sequencing libraries using the Nextera© XT Sample Preparation Kit and the Nextera© XT Index Kit as recommended by the manufacturer. The libraries were sequenced on a MiSeq instrument with the MiSeq reagent kit version 3 as recommended by the manufacturer (Illumina, San Diego, CA, USA). Trimmomatic version 0.36 was used for quality-filtering of the raw data [21]. The number of paired-end reads after filtering were: 2,605,156 (DVP1-17-2344); 2,783,864 (DVP4-16-6166); 2,708,816 (14T0637); 2,286,452 (13T098); 1,975,930 (DVP1-17-1678). The assemblies were performed with the SPAdes genome assembler software (version 3.11.1) [22]. QualiMap v.2.2.1 was used to validate the assemblies [23]. The assemblies resulted in a coverage of the genomes between 200- and 291-fold. The contig numbers were between 10 and 13. Further information regarding the genome sequences and all GenBank accession numbers of the newly and previously sequenced genomes are listed in Table 3.

### 4.6. Bioinformatics and Phylogenetic Analyses

Gene prediction and annotation of all genomes were done with NCBI Prokaryotic Genome Annotation Pipeline (PGAP) [24]. For phylogenomic analyses, the core genome was identified and aligned with Parsnp, a program that is part of the Harvest software package [25]. Only reliable core-genome single-nucleotide polymorphisms (SNPs) were considered for reconstruction of the core-genome phylogeny. Phylogenetic trees were visualized using Mega v7 [26] and Interactive Tree Of Life (iTOL, version 5.7) [27]. Genome comparisons and visualization were done with the Artemis Comparison Tool ACT [28] and with BRIG [29]. BlastKOALA (v2.2), an internal annotation tool of KEGG (Kyoto Encyclopedia of Genes and Genomes), was used to map genes to metabolic pathways [30].

## Figures and Tables

**Figure 1 pathogens-10-00397-f001:**
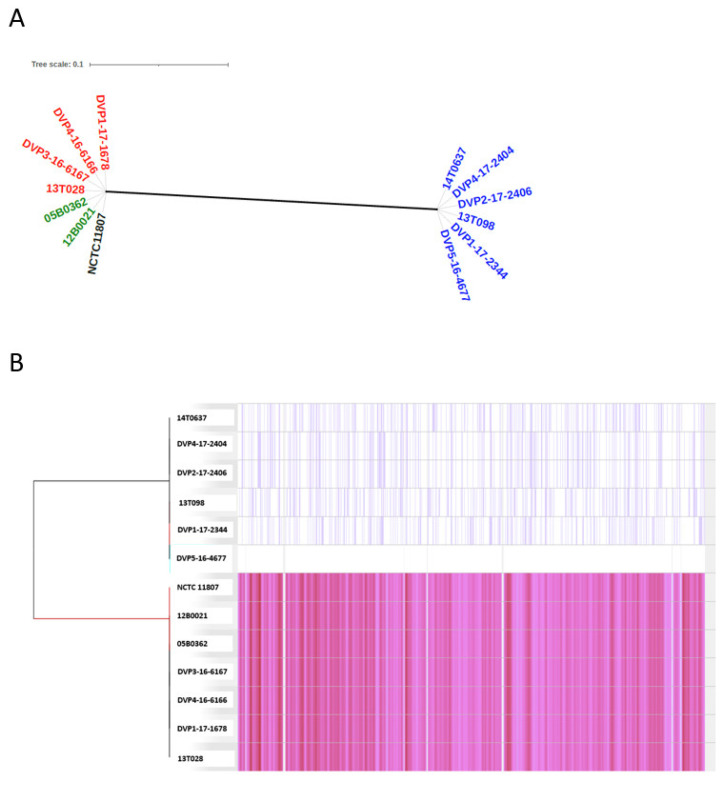
Phylogeny of the *Staphylococcus saccharolyticus* population, based on core-genome comparison. Available *S. saccharolyticus* genomes were compared, including previously sequenced genomes [3] and newly sequenced genomes (five strains). The core genome was identified; it covers 93% of the reference genome (here: the closed genome of strain DVP5-16-4677). (**A**) A phylogenetic reconstruction based on core-genome SNPs reveals that the strains can be divided into two clades. In blue, strains isolated from hip PJIs; in red, strains isolated from shoulder PJIs; in green, blood culture isolates; in black, the type strain of the species, NCTC 11807 (GenBank accession number: UHDZ00000000). (**B**) SNP locations within the core genome. Strain DVP5-16-4677 was used as reference. A single line represents a SNP; grey regions represent non-core-genome regions. In total, 34,107 SNPs were detected. Parsnp was used for this analysis and visualization was done in Gingr.

**Figure 2 pathogens-10-00397-f002:**
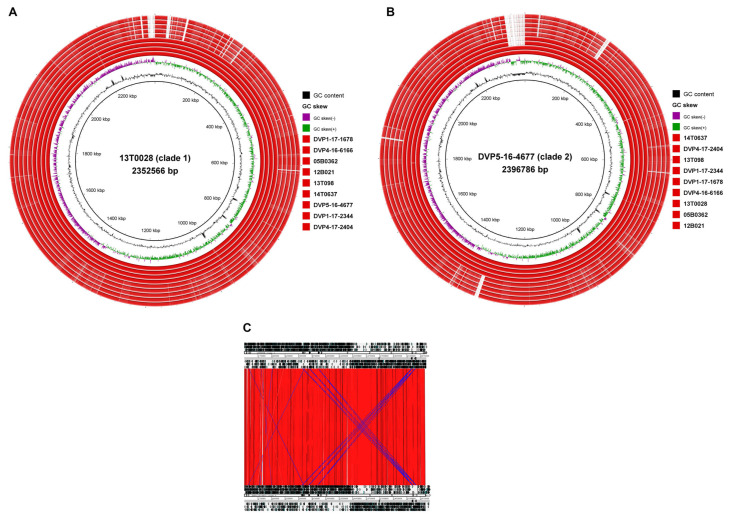
Genome-wide synteny in *S. saccharolyticus.* Five clade 1 strains and five clade 2 strains were compared, revealing a strong synteny, with overall few clade-specific regions. (**A**). The genome of clade 1 strain 13T028 was compared by Blast against other genomes of clade 1 and clade 2 strains of *S. saccharolyticus*. Two clade 1-specific islands >8 kb could be detected (Appendix A). (**B**). The genome of clade 2 strain DVP5-16-4677 was compared against other clade 2 and clade 1 *S. saccharolyticus* genomes. Three clade 2-specific islands >8 kb could be detected (Appendix A). Visualization was done with BRIG. (**C**). The ACT tool was used to directly compare the closed genomes of strains 13T028 (clade 1, top) and DVP5-16-4677 (clade 2, bottom) by Blast. Strong genome synteny was observed; no large insertions or inversions could be detected.

**Figure 3 pathogens-10-00397-f003:**
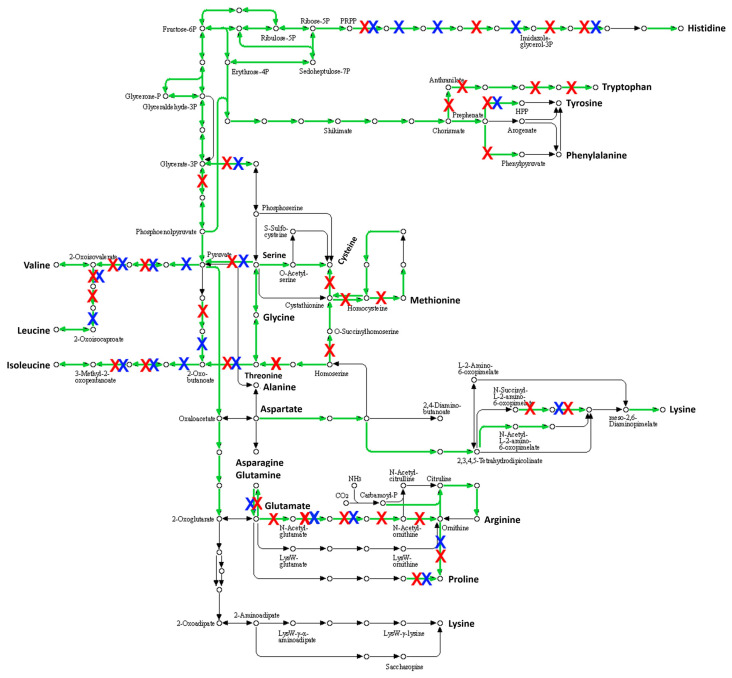
Mutational inactivation of many genes involved in amino acid biosynthesis in *S.*
*saccharolyticus*. Shown are pathways involved in amino acid biosynthesis (KEGG map01230). BlastKOALA was used to map gene products to metabolic pathways. Green represents identified gene products (including pseudogenes) found in the genomes of DVP5-16-4677 and 13T028 that can catalyze these reactions according to BlastKOALA. Red and blue crosses represent pseudogenes; red cross: frameshift mutation in the respective gene in strain DVP5-16-4677; blue cross: frameshift mutation in the respective gene in strain 13T028.

**Table 1 pathogens-10-00397-t001:** Clinical data for seven patients with orthopedic implant-associated infections due to *Staphylococcus saccharolyticus*.

Pat	Patient ID Sex, Age	Isolate ID	Affected Joint	Characteristics (Diagnosis, Surgical Procedure)	Time to Diagnosis (from Primary Surgery)	Duration of Symptoms	Type of Infection (Number of Tissue Biopsies with Growth)	Antimicrobial Treatment, Duration	Surgical Procedure and Follow-Up
1	♂, 76	DVP2-17- 2406	Hip	Osteoarthritis. Primary arthroplasty surgery 2002 (Exeter). Loosening of the cup in 2015.	13 years	Pain 2 years, affected functional status	Monomicrobial *Staphylococcus saccharolyticus*, 2/5 positive cultures	No antibiotics administered	One-stage exchange procedure of cup and femoral head due to loosening. Two years later good functional status, no walking aid.
2	♂, 65	DVP5-16-4677	Hip	Osteoarthritis. Primary arthroplasty surgery May 2014 (Stryker). Loosening of the cup in July 2015.	14 Months	Pain 1 year, affected functional status	Monomicrobial *Staphylococcus saccharolyticus*, 3/5 positive cultures	Amoxicillin, 4 months	One-stage exchange procedure of cup and femoral head due to loosening. One year later good functional status, no walking aid.
3	♂, 76	14T637	Hip	Osteoarthritis. Primary arthroplasty surgery 1998 (Exeter). Loosening of the cup in 2014.	16 years	Pain ca 6 months, affected functional status	Polymicrobial *Cutibacterium acnes* 4/5, *Staphylococcus saccharolyticus* 1/5	Amoxicillin, 3 months	One-stage exchange procedure of cup due to loosening. One year later acceptable functional status, walking aids sporadically.
4	♂, 69	13T098	Hip	Osteoarthritis. Primary arthroplasty surgery 2005 (Exeter). Loosening of the cup in 2012.	7 years	Pain ca 1 year	Monomicrobial *Staphylococcus saccharolyticus*, 3/5 positive cultures	No antibiotics administered	One-stage exchange procedure of cup and stem due to loosening. One year later excellent functional status, no walking aid.
5	♂, 75	DVP1-17-2344	Hip	Osteoarthritis. Primary arthroplasty surgery 2001 (Lubinus). Loosening of the cup in 2016.	15 years	Pain ca 6 moths	Polymicrobial *Staphylococcus saccharolyticus* 3/5, *Cutibacterium acnes* 2/5,	Amoxicillin, 3 months	One-stage exchange procedure of all prosthetic devices due to loosening. One year later excellent functional status, no walking aid.
6	♂, 72	13T028	Shoulder	Traumatic fracture of proximal humerus. Global FX Shoulder Prosthesis 2012. Early post-operative infection.	21 days	Pain, swelling, local inflammation 17 days post-opertively	Polymicrobial *Staphylococcus saccharolyticus* 5/5, *Cutibacterium acnes* 4/5	Penicillin V, 4 months	Revision and lavage 1 month post-operatively. One year later poor functional status, ROM 30°.
7	♂, 49	DVP4-16-6166, DVP1-17-1678	Shoulder	Secondary osteoarthritis due to *Staphylococcus aureus* osteomyelitis. Tissue biopsies at primary arthroplasty surgery (Global Unite) showed growth of *Staphylococcus saccharolyticus 2*/*6*, *Cutibacterium acnes* 1/6	5 months	Pain, swelling, local inflammation post-operatively	Polymicrobial *Staphylococcus saccharolyticus* 3/5, *Cutibacterium acnes* 1/5	Penicillin V, 3 months	Two-stage exchange procedure. Antibiotic treatment; 14 months follow-up shows good functional status.

♂ = male sex.

**Table 2 pathogens-10-00397-t002:** Minimum inhibitory concentration (MIC) values (mg/L) for three antibiotics of *Staphylococcus saccharolyticus* isolates associated with prosthetic joint infections (*n* = 7).

Patient Lab ID	1 DVP2-17-2406	2 DVP5-16-4677	3 14T637	4 13T098	5 DVP1-17-2344	6 13T028	7 DVP4-16-6166
Benzyl-penicillin	0.002	<0.002	0.004	0.003	<0.002	0.003	0.002
Clindamycin	0.016	0.032	0.032	0.032	0.032	0.047	0.125
Metronidazole	>256	>256	>256	>256	>256	>256	>256
Amoxicillin	<0.016	<0.016	<0.016	<0.016	<0.016	<0.016	<0.016
Imipenem	<0.008	<0.004	<0.008	<0.016	<0.008	<0.004	<0.004
Meropenem	<0.002	<0.002	<0.002	<0.002	<0.002	<0.002	<0.002
Vancomycin	0.5	1.0	1.0	1.0	1.0	1.0	1.0
Chloramphenicol	4	1	1	2	2	0.125	0.16

**Table 3 pathogens-10-00397-t003:** Genomic characteristics of *Staphylococcus*
*saccharolyticus* strains associated with prosthetic joint infections.

Strain	Source	Genome Size (kb)	G+C (%)	Contigs	Coverage	N50 (kb)	Clade	CDS	Pseudogenes	Accession Number
DVP2-17-2406	Patient 1, hip	2373	32.00	11	222	1222	2	1740	502	QHKD00000000
DVP5-16-4677	Patient 2, hip	2396	32.08	2	188	-	2	1726	493	Chromosome: CP068031.1 Plasmid: CP068032.1
**14T637**	Patient 3, hip	2375	32.00	13	257	521	2	1732	487	JAENGT000000000
**13T098**	Patient 4, hip	2321	32.10	10	225	1224	2	1704	485	JAENGW000000000
**DVP1-17-2344**	Patient 5, hip	2321	32.10	10	254	1223	2	1711	485	JAENGU000000000
13T028	Patient 6, shoulder	2352	32.16	2	205	-	1	1750	469	Chromosome: CP068029.1 Plasmid: CP068030.1
**DVP4-16-6166**	Patient 7, shoulder	2349	32.00	13	291	749	1	1778	470	JAENGX000000000
**DVP1-17-1678**	Patient 7, shoulder	2349	32.00	10	200	768	1	1775	471	JAENGV000000000

At least one strain of each of the seven patients was sequenced; newly sequenced isolates are marked in bold.

## Data Availability

The five new draft genome sequences reported in this study are available at GenBank with the accession numbers: JAENGT000000000 (strain 14T0637), JAENGU000000000 (strain DVP1-17-2344), JAENGV000000000 (strain DVP1-17-1678), JAENGW000000000 (strain 13T098), JAENGX000000000 (strain DVP4-16-6166).

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
