# Peer review of "Staphylococcus saccharolyticus Associated with Prosthetic Joint Infections: Clinical Features and Genomic Characteristics"

_pathogens, 2021, doi:10.3390/pathogens10040397_

Round 1
Reviewer 1 Report
This is a research article focused on the analysis of Staphylococcus saccharolyticus isolates from prosthetic joint infections. Although the data are very novel due to the low level of information related to this particular bacterium in human diseases, the report is quite preliminary. Several sections should be improved:
-The antibiotic susceptibility testing is minimal. If authors aim to characterize the antimicrobial resistance of this pathogen, the list of antibiotics tested must expand. The authors should add here antibiotics recommended by EUCAST for Gram-positive anaerobic bacteria. I.e. amoxicillin, imipenem, meropenem, vancomycin, and chloramphenicol.
-The analysis of the genomes is also minimal. Authors should expand this to produce figures on the genomes' synteny to show the differences in the core genome and the regions with high variability. This is particularly important because there is a link between genome sequence and the isolates' origin (lines 150-151). Authors should also analyze the metabolism of S. saccharolyticus, which could be done quickly with software such as Pathway Tools. This is relevant considering that the genome decay and the adaptation of the pathogen to the human environment since the particular metabolic traits of S. saccharolyticus may shed light on the biology behind the colonization of prosthetic joints.
-The lack of mutations in the genome sequence of strains isolated on the same day or four months apart is considered remarkable in the article. Did the authors expect changes in the genome sequence of strains isolated on the same day? If so, I assume that this is due to the relatively high number of pseudogenes in the genomes, and therefore authors should link both observations. However, there could be an alternative explanation for this. The rationale behind the sequencing effort of these isolates should be explained in more detail. For example, what do authors consider the core genome in this context? Why did the authors focus on the core genome for this comparison? Did other regions of the genome change substantially?
-Authors should expand the discussion with a section on any potential preventative strategies that may be useful to implement in the production and implantation of prosthetic joints.
Author Response
This is a research article focused on the analysis of Staphylococcus saccharolyticus isolates from prosthetic joint infections. Although the data are very novel due to the low level of information related to this particular bacterium in human diseases, the report is quite preliminary. Several sections should be improved:
-The antibiotic susceptibility testing is minimal. If authors aim to characterize the antimicrobial resistance of this pathogen, the list of antibiotics tested must expand. The authors should add here antibiotics recommended by EUCAST for Gram-positive anaerobic bacteria. I.e. amoxicillin, imipenem, meropenem, vancomycin, and chloramphenicol.
AST results of the requested antibiotics have been added
-The analysis of the genomes is also minimal. Authors should expand this to produce figures on the genomes' synteny to show the differences in the core genome and the regions with high variability. This is particularly important because there is a link between genome sequence and the isolates' origin (lines 150-151). Authors should also analyze the metabolism of S. saccharolyticus, which could be done quickly with software such as Pathway Tools. This is relevant considering that the genome decay and the adaptation of the pathogen to the human environment since the particular metabolic traits of S. saccharolyticus may shed light on the biology behind the colonization of prosthetic joints.
A genome synteny figure is now included as a new figure (Fig. 2) in the main text. We included also information regarding the accessory genome, including a list of genome differences between clade 1 and clade 2 strains as supplement table (Table S1). The results part was extended to cover these results.
The metabolism was analyzed with KEGG/BlastKOALA. We added parts of the outcome in the results part and in the discussion, in particular regarding the pathways that seem to be affected by the genome decay (Fig 3, Table S2). It is beyond the scope of this article to perform an in-depth metabolic reconstruction of S. saccharolyticus.
-The lack of mutations in the genome sequence of strains isolated on the same day or four months apart is considered remarkable in the article. Did the authors expect changes in the genome sequence of strains isolated on the same day? If so, I assume that this is due to the relatively high number of pseudogenes in the genomes, and therefore authors should link both observations. However, there could be an alternative explanation for this. The rationale behind the sequencing effort of these isolates should be explained in more detail. For example, what do authors consider the core genome in this context? Why did the authors focus on the core genome for this comparison? Did other regions of the genome change substantially?
No, we did not expect any changes among strains isolated on the same day. We now omitted this information regarding the strains isolated on the same day. However, we were surprised not to see genomic changes when strains were isolated four months apart, given the human defense systems at play and potential bacterial evasion strategies that could lead to mutations/SNPs (and, mentioned by the reviewer, given the high number of existing pseudogenes, we expected ongoing changes). We now compared the whole genome (core and accessory genome) of the two strains that were isolated four months apart and did not detect any changes.
Regarding the core and accessory genome, we added some information, e.g. differences of the accessory genome between strains (of different clades) (Fig. 2, Table S1). The accessory genome is rather small. Regarding the core genome: it was calculated with Parsnp and clade-/strain-specific SNPs in the core genome are shown in Fig. 1B and Fig. S1.
-Authors should expand the discussion with a section on any potential preventative strategies that may be useful to implement in the production and implantation of prosthetic joints.
Two sentences regarding antibiotic prophylaxis has been added in the discussion section.
Reviewer 2 Report
This manuscript by Bo Söderquist et al. describes some very interesting data on S. saccharolyticus. Global : Bacterial names, "e.g", "i.e." must be in italics. Numbers less than 12 must be written in full letters. Results: Could the authors give the duration of symptoms for the first three cases (Table 1)? CRP assays should be detailed (name of manufacturer, location) (line 106) Methods: A flowchart would be very helpful in understanding the process of selecting the patients to be analyzed. How were the five sequenced strains selected? Did the authors consider verifying the identification by MALDI-TOF of patients collected before January 2014? EUCAST guidelines should be referenced according to the recommendations for authors. What were the validation criteria after observing the 1% agarose gel? Did the authors consider a minimum concentration at the generated shotgun libraries? Could the interactive tree of life be referenced? What is the version used by the authors?
Author Response
Reviewer #2
This manuscript by Bo Söderquist et al. describes some very interesting data on S. saccharolyticus.
Global : Bacterial names, "e.g", "i.e." must be in italics.
e.g. and i.e. are now in italics
Numbers less than 12 must be written in full letters.
Has been corrected
Results:
Could the authors give the duration of symptoms for the first three cases (Table 1)?
Sorry for that, clinical data have been added in the Table.
CRP assays should be detailed (name of manufacturer, location) (line 106)
Manufacturer of the CRP instrument for routine analysis of CRP at the Department of Laboratory Medicine, Clinical Chemistry; Örebro University Hospital have been added in M&M, point 4.4. Questionable if this information is relevant. Can be deleted.
Methods: A flowchart would be very helpful in understanding the process of selecting the patients to be analyzed.
Since all but one isolates were determined to species level by API or MALDI-TOF MS we do not find it relevant to show a flow chart for 7 patients. All isolates were verified to species level with MALDI-TOF MS databases 7311 and 7854.
How were the five sequenced strains selected?
To make sure that at least one isolate of each patient was sequenced.
Did the authors consider verifying the identification by MALDI-TOF of patients collected before January 2014?
As mentioned in the text of version 1 of the manuscript “All isolates were retrospectively assigned to species level by MALDI-TOF MS.”
EUCAST guidelines should be referenced according to the recommendations for authors.
Have been revised.
What were the validation criteria after observing the 1% agarose gel?
The criteria are now specified in the revised version of the manuscript (methods section).
“The purity and quality of the gDNA were assessed on a 1% agarose gel and with a nanodrop apparatus (Thermo Fisher Scientific). The validation criteria were: sample quality 260/280nm = 1.8-2 and 260/230nm = 2.0-2.2 and a minimum concentration 2.5 µg/µl.”
Did the authors consider a minimum concentration at the generated shotgun libraries?
Yes, see DNA input concentration specified above (>2.5 µg/µl). In addition, we included the total number of paired-end reads generated in the methods section. All five new genomes were sequenced with a minimum coverage of 200fold (table 3)
Could the interactive tree of life be referenced? What is the version used by the authors?
We included a reference and mentioned the version number in the methods section (Version number 5.7). Reference: Interactive Tree Of Life (iTOL) v4: recent updates and new developments. Letunic I, Bork P. Nucleic Acids Res. 2019 Jul 2;47(W1):W256-W259.
Reviewer 3 Report
Söderquist et al describe clinical features of patients infected with Staphylococcus saccharolyticus, an organism rarely found in clinical specimens. The authors then perform whole genome sequencing on 5 of these isolates to determine potential virulence traits associated with pathogenicity. By combining with other previously sequenced isolates the authors conclude that there are two clades based on SNP analysis of the core genome.
The main outcome of this manuscript is that there are two phylogenetic clades associated with specific body sites/infections within S saccharolyticus with little strain individuality among strains from the same clade. However, this reviewer would argue that the very low sample size and the lack of genetic or phenotypic evidence suggesting niche-specific adaption challenges the conclusion that there are two clades that are associated with different body sites, hip PJIs (Clade 2) and shoulder PJIs (Clade 1). Moreover, in clade 2 three of these strains are from the same patient which significantly impacts strain individuality. Furthermore, there is no data suggesting how genetic variability between the different clades impacts virulence and no specific virulence traits have been identified as suggested in the abstract. Therefore, this narrative should be changed. A comparison between the genomes of commensal S saccharolyticus and strains isolated from infection would potentially aid in identifying virulence signatures.
At a minimum, I would suggest that the authors indicate where the SNPs are occurring in genes, highlighting any virulence/metabolic/antibiotic/regulatory genes, backing up their claims of niche-specific adaption to certain skin sites of the two clades.
The manner in which the manuscript is written is confusing and uses isolates previously sequenced and published in other journals. For example line 67 the authors state they identified 5 patients, but provide clinical data for 7 patients (Table1). Table 3 notes genomic characteristics of newly (n=5) and previously sequenced (n=5) genomes of S. saccharolyticushowever only 7 strains are provided in Table 3. In addition multiple strains from the same patient has been sequenced and included in the phylogenetic tree in Fig 1. The strain and patient descriptions should be made clearer, noting strains identified and sequenced in this study and those in other publications.
Author Response
Reviewer #3
Söderquist et al describe clinical features of patients infected with Staphylococcus saccharolyticus, an organism rarely found in clinical specimens. The authors then perform whole genome sequencing on 5 of these isolates to determine potential virulence traits associated with pathogenicity. By combining with other previously sequenced isolates the authors conclude that there are two clades based on SNP analysis of the core genome.
The main outcome of this manuscript is that there are two phylogenetic clades associated with specific body sites/infections within S saccharolyticus with little strain individuality among strains from the same clade. However, this reviewer would argue that the very low sample size and the lack of genetic or phenotypic evidence suggesting niche-specific adaption challenges the conclusion that there are two clades that are associated with different body sites, hip PJIs (Clade 2) and shoulder PJIs (Clade 1). Moreover, in clade 2 three of these strains are from the same patient which significantly impacts strain individuality. Furthermore, there is no data suggesting how genetic variability between the different clades impacts virulence and no specific virulence traits have been identified as suggested in the abstract. Therefore, this narrative should be changed. A comparison between the genomes of commensal S saccharolyticus and strains isolated from infection would potentially aid in identifying virulence signatures.
We agree that a low sample size limits the conclusion and we changed the narrative accordingly in the results and discussion part. We added more information regarding genome differences of clade 1 and clade 2 strains (accessory and core genome differences, see answer to reviewer 1). Thank you for the suggestion regarding future work to investigate differences between skin-resident S. saccharolyticus strains and PJI-associated strains (at the moment there is no genome information available on skin-resident S. saccharolyticus)
At a minimum, I would suggest that the authors indicate where the SNPs are occurring in genes, highlighting any virulence/metabolic/antibiotic/regulatory genes, backing up their claims of niche-specific adaption to certain skin sites of the two clades.
We added data regarding core and accessory genome differences between clade 1 and clade 2 strains in the results part. Regarding SNPs in the core genome: Parsnp predicted a core genome of the species that covers 93% from the closed reference genome (Fig 1B). In total, 34,107 SNPs were detected. When just comparing the (seven) clade 1 strains, 298 SNPs were detected in the core genome (covering 93% of the genome, strain 13T0028 as reference), and when just comparing the (six) clade 2 strains, 645 SNPs were detected in the core genome (DVP5-16-4677 as reference). This means that clade 1 and clade 2 strains differed by around 33,000 SNPs; these were distributed throughout the genome and affected all genes (see also Fig 1B and text). We had a closer look on the identified pseudogenes. BlastKOALA was used to map gene products to (metabolic) pathways and predict which pathways were affected by mutational inactivation (table S2 and text in results and discussion).
The manner in which the manuscript is written is confusing and uses isolates previously sequenced and published in other journals. For example line 67 the authors state they identified 5 patients, but provide clinical data for 7 patients (Table1). Table 3 notes genomic characteristics of newly (n=5) and previously sequenced (n=5) genomes of S. saccharolyticus however only 7 strains are provided in Table 3. In addition multiple strains from the same patient has been sequenced and included in the phylogenetic tree in Fig 1. The strain and patient descriptions should be made clearer, noting strains identified and sequenced in this study and those in other publications.
We reorganized the manuscript accordingly. We think that the patients (n=7) and the selected strains (n=8; one strain from each patient 1-6, and 2 strains from patient 7) are now presented more clearly. Table 3 is now updated. Please see also the comment to reviewer 1 regarding strain choice.
Round 2
Reviewer 1 Report
I do not have any further comments.
Reviewer 3 Report
The authors have addressed all my comments.